# Progression Risk Score Estimation Based on Immunostaining Data in Oral Cancer Using Unsupervised Hierarchical Clustering Analysis: A Retrospective Study in Taiwan

**DOI:** 10.3390/jpm11090908

**Published:** 2021-09-13

**Authors:** Hui-Ching Wang, Leong-Perng Chan, Chun-Chieh Wu, Hui-Hua Hsiao, Yi-Chang Liu, Shih-Feng Cho, Jeng-Shiun Du, Ta-Chih Liu, Cheng-Hong Yang, Mei-Ren Pan, Sin-Hua Moi

**Affiliations:** 1Graduate Institute of Clinical Medicine, College of Medicine, Kaohsiung Medical University, Kaohsiung 807, Taiwan; joellewang66@gmail.com (H.-C.W.); ashiun@gmail.com (J.-S.D.); mrpan@cc.kmu.edu.tw (M.-R.P.); 2Department of Internal Medicine, Division of Hematology and Oncology, Kaohsiung Medical University Hospital, Kaohsiung Medical University, Kaohsiung 807, Taiwan; huhuhs@cc.kmu.edu.tw (H.-H.H.); ycliu@kmu.edu.tw (Y.-C.L.); sfcho@kmu.edu.tw (S.-F.C.); 3Faculty of Medicine, College of Medicine, Kaohsiung Medical University, Kaohsiung 807, Taiwan; oleon24@yahoo.com.tw; 4Department of Otolaryngology-Head and Neck Surgery, Kaohsiung Medical University Hospital, Kaohsiung Medical University, Kaohsiung 807, Taiwan; 5Department of Otorhinolaryngology-Head and Neck Surgery, Kaohsiung Municipal Ta-Tung Hospital and Kaohsiung Medical University Hospital, Kaohsiung 807, Taiwan; 6Department of Pathology, Kaohsiung Medical University Hospital, Kaohsiung Medical University, Kaohsiung 807, Taiwan; 930220@kmuh.org.tw; 7Department of Hematology-Oncology, Chang Bing Show Chwan Memorial Hospital, Changhua 505, Taiwan; touchyou3636@gmail.com; 8Department of Electronic Engineering, National Kaohsiung University of Science and Technology, Kaohsiung 807, Taiwan; chyang@nkust.edu.tw; 9Ph.D. Program in Biomedical Engineering, Kaohsiung Medical University, Kaohsiung 807, Taiwan; 10Drug Development and Value Creation Research Center, Kaohsiung Medical University, Kaohsiung 807, Taiwan; 11Center of Cancer Program Development, E-Da Cancer Hospital, I-Shou University, Kaohsiung 807, Taiwan

**Keywords:** oral cancer, risk stratification, progression-free survival

## Abstract

This study aimed to investigate whether the progression risk score (PRS) developed from cytoplasmic immunohistochemistry (IHC) biomarkers is available and applicable for assessing risk and prognosis in oral cancer patients. Participants in this retrospective case-control study were diagnosed between 2012 and 2014 and subsequently underwent surgical intervention. The specimens from surgery were stained by IHC for 16 cytoplasmic target markers. We evaluated the results of IHC staining, clinical and pathological features, progression-free survival (PFS), and overall survival (OS) of 102 oral cancer patients using a novel estimation approach with unsupervised hierarchical clustering analysis. Patients were stratified into high-risk (52) and low-risk (50) groups, according to their PRS; a metric consisting of cytoplasmic PLK1, PhosphoMet, SGK2, and SHC1 expression. Moreover, PRS could be extended for use in the Cox proportional hazard regression model to estimate survival outcomes with associated clinical parameters. Our study findings revealed that the high-risk patients had a significantly increased risk in cancer progression compared with low-risk patients (hazard ratio (HR) = 2.20, 95% confidence interval (CI) = 1.10–2.42, *p* = 0.026). After considering the influences of demographics, risk behaviors, and tumor characteristics, risk estimation with PRS provided distinct PFS groups for patients with oral cancer (*p* = 0.017, *p* = 0.019, and *p* = 0.020). Our findings support that PRS could serve as an ideal biomarker for clinical use in risk stratification and progression assessment in oral cancer.

## 1. Introduction

A high occurrence of oral cancer recurrence and metastasis events result from this cancer’s late presentation, resulting in poor survival in patients with oral cancer [1]. Multidisciplinary interventions, including radical surgery, radiotherapy, and cytotoxic chemotherapy, worsen the quality of life of patients. Thus, the creation of a practical approach to the interaction among clinicopathologic factors, immunohistochemistry, and genetic specificity to estimate the outcomes and prognosis of patients has been gradually emphasized [2]. Early diagnosis and identification of high-risk patients for potential recurrence prevent their progression and improve their survival.

In oral cancer, well-known clinicopathologic factors, such as tumor size, stage, nodal status, the positivity of margin, lymphovascular invasion, perineural invasion, and extranodal extension, are widely regarded as having potential for risk stratification for further therapeutic strategies [3,4]. Some studies have evaluated chromogen-based in situ hybridization (ISH) and immunohistochemistry (IHC) biomarkers from cancerous tissues or databases, such as stromal microRNA-204 and RFC4, to assess the feasibility of prognostic prediction [5,6]. Recent studies have started to emphasize personalized biomarker-driven therapeutic strategies to guide treatments in refractory advanced cancer, including basket trials and umbrella trials [7,8]. Some biomarker-driven treatment strategies have even moved to guide multidisciplinary interventions. For example, the epithelial-mesenchymal transcription marker Slug predicts the survival benefit of up-front surgical intervention for head and neck cancer. Patients with high Slug expression on IHC have a higher risk of radio- and chemotherapy resistance, and earlier surgery resulted in better survival than either definitive radiotherapy or chemoradiotherapy [9]. However, there is still a lack of standard guidelines for biomarkers of IHC expression or genetic alterations to predict the treatment response or prognosis in oral cancer.

The cytoplasm consists of the cytosol and organelles. Antibodies for biomarkers detect proteins within the cytoplasm, which can modulate cell morphology and cytoskeletal structure. Cytoplasmic markers can clarify the specific roles of a protein and illustrate the executive tasks of a protein in cancer cells. With the consequent explosion of various genomic and molecular data, an upcoming question is how to organize the high-throughput clinical data into meaningful interpretations and structures. Unsupervised hierarchical clustering analysis has been widely used to separate biological objects with common characteristics into different groups and to integrate data by underlying biology. In non-small cell lung carcinoma, unsupervised hierarchical clustering analysis successfully identified and stratified different subgroups of patients based on molecular expression profiles [10]. In breast cancer, hierarchical clustering analysis demonstrated that the overexpression of hypomethylated X-linked genes was associated with lower survival rates [11]. However, the clinical applications of clustering algorithms are insufficient in cancer patients. In this study, we aimed to develop a novel approach and calculation to evaluate prognostic biomarkers according to the diverse expression of cytoplasmic IHC staining.

## 2. Materials and Methods

### 2.1. Patient Selection

We collected 163 patients with oral cavity cancers from the Kaohsiung Medical University Hospital and followed up these patients for 5 years. We included the patients based on the following criteria: patients older than 20 years old, ICD-9 site code specific for the oral cavity, squamous cell carcinoma with a histologic grading of 1 to 3 (well-differentiated, moderately differentiated, and poorly differentiated type), patients who underwent wide excision, and diagnosis between 2012 and 2014. The exclusion criteria included patients who underwent biopsy without wide excision, with secondary malignancy, histology of carcinoma in situ, and SCC of the nasopharynx, oropharynx, hypopharynx, and larynx. We retrospectively collected medical records, including age, sex, areca nut usage, alcohol consumption, tobacco habits, and other clinical parameters. The clinicopathological factors we recorded included histologic type and grade, tumor size, lymph node status, surgical margin, perineural invasion (PNI), lymphovascular invasion (LVI), and extranodal extension (ENE). We excluded patients without complete clinical data and clinicopathological factors. Finally, 102 patients were analyzed. We evaluated the results of a retrospective study with the primary endpoint of assessing outcomes at a comprehensive cancer institution in southern Taiwan. We analyzed progression-free survival (PFS) and overall survival (OS) after surgery. This study was approved by the Institutional Review Board and Ethics Committee of Kaohsiung Medical University Hospital (KMUHIRB-E(I)-20170034). The data were analyzed anonymously; therefore, no informed consent was obtained. All methods were performed under approved guidelines and regulations.

### 2.2. Tissue Microarrays and Immunostaining

We adopted an analysis similar to that used in our previous study to identify novel IHC prognostic biomarkers associated with synthetic lethality (SL) in lung adenocarcinoma and colorectal cancer [12,13]. The SL-associated genes included oncogenes, tumor suppressor genes, and genome stability genes. From these validated SL-associated genes, we selected 16 genes to perform cytoplasmic IHC staining and evaluated the possible cytoplasmic IHC prognostic markers among them.

Figure 1 illustrates the schematic diagram for target gene selection from the vali-dated SL gene pairs and the identification of the protein staining matrix according to the 16 individual cytoplasmic IHCs. Initially, 742 SL pairs of genes were selected, and the microarray gene expression data from the Cancer Genome Atlas (TCGA) of 79 Asian OSCC samples (57 cancerous and 22 noncancerous) were analyzed. Gene expression datasets were screened according to the following parameters: cancerous and noncancerous tissues, no treatments, no metastasis, and Affymetrix chips (up to November 2010). OSCC genes were downloaded from the GEO database [14]. Gene expression data were collected from patients of Han Chinese origin (57 OSCC and 22 noncancerous tissues from Taiwanese patients, GSE 25099), the same ethnicity as that of the IHC and clinicopathological data previously used [15]. Gene expression profiles for the 57 OSCC and 22 noncancerous tissues in the dataset were quantile-normalized using the “expresso” function in R, and log ratios were computed for the target gene expression in each cancerous tissue versus the mean expression in the noncancerous tissues. The selected SL gene pairs were further sorted by the fractions of the upregulation and downregulation patterns, and the SL pairs with 1.5-fold differential expression in fractions computed from gene pairs were selected as target genes. Overall, 21 genes were selected using the above criteria, and the cancer specimens collected from the Taiwanese population in the current study were then used to produce tissue microarrays with three cancerous and one noncancerous tissue core, as in our previous study [16]. Tissue microarrays were further processed for the cytoplasmic IHC of 16 target proteins among the 21 genes. Hence, 16 protein staining scores were obtained for the 16 target proteins. The 16 target cytoplasmic proteins included FEN1, FLNA, PIM1, STK17A, CDH3, SHC1, POLB, SGK2, PhosphoMet, CNSK1E, PLK1, CDK6, KRAS, EGFR, RB1, and P16. The antibodies and retrieval buffers for each protein are summarized in Table 1. In addition, the cancer tissue samples from two OSCC patients with IHC staining using control IgG antibody are summarized in Appendix A.

### 2.3. Data Analysis

The baseline characteristics of the study population according to PFS status are summarized in terms of frequency and percentage. Two survival outcomes were observed: PFS and OS. For PFS, the patients who were diagnosed with progressive disease within the study follow-up period were defined as disease-progressed cases; otherwise, they were defined as progression-free cases. For OS, the patients who died within the study follow-up period were defined as dead cases, and the remaining patients were defined as alive cases. Both survival outcomes were tracked from the first diagnosis date of oral cancer until the date of disease progression or death, while the disease-free and alive cases were tracked until the last date of study follow-up.

Unsupervised hierarchical clustering was used to identify the protein combinations according to the similarity of the immunostaining profiles. The unsupervised hierarchical clustering analysis workflow for the protein staining results is shown in Figure 2. First, the staining intensity of the 16 target proteins was transformed into a normalized staining matrix. Subsequently, each protein was assigned to the corresponding cluster to generate a protein cluster, and the optimal number of clusters was determined using the silhouette index. Next, the patients were dichotomized into two groups according to the immunostaining profiles of each protein cluster. Therefore, the group with a higher proportion of disease progression was defined as the high-risk group; otherwise, it was defined as the low-risk group. Therefore, the survival difference between the high-risk and low-risk groups in each protein cluster was estimated using the log-rank test.

The protein cluster with a significant survival difference between the high-risk and low-risk groups was selected as a candidate cluster. Therefore, the candidate protein cluster was used to derive a PRS for oral cancer PFS and OS. The calculation of PRS was as follows:(1)$$S_{h}=\|P_{1}-{\bar{H}}_{1}\|+\|P_{2}-{\bar{H}}_{2}\|+\cdots +\|P_{n}-{\bar{H}}_{n}\|$$
where P is the immunostaining score for a specific target protein and $\bar{H}$ is the harmonic mean of the immunostaining score in the high-risk group.
(2)$$S_{l}=\|P_{1}-{\bar{L}}_{1}\|+\|P_{2}-{\bar{L}}_{2}\|+\|P_{2}-{\bar{L}}_{2}\|$$
where P is the immunostaining score for a specific target protein and $\bar{\bar{L}}$ is the harmonic mean of the immunostaining score in the low-risk group.
(3)$$PRS=S_{h}-S_{l}$$
where $S_{h}$ is computed from Equation (1) and $S_{l}$ is computed from Equation (2).

A positive PRS value indicates that the patient might increase the risk of disease progression, while a negative PRS indicates the opposite situation. Therefore, the PRS was used to determine the appropriate risk strata for the study population. The Kaplan–Meier method was used to compare the survival curves between PRS risk strata. The Cox proportional hazard regression model was used to evaluate the impact of PRS risk strata on PFS and OS. The multiple model comparisons for PFS and OS, including PRS risk strata, were illustrated using forest plots. All *p*-values were two-sided, and the significance level was set at 0.05. All analyses were performed using the computing environment R 4.0.2 (R Core Team, 2020).

## 3. Results

### 3.1. Baseline Characteristics

A total of 102 patients were retrospectively analyzed. The baseline characteristics of the study population according to PFS status are summarized in Table 2. Sixty-six patients were maintained in a progression-free status, while 36 patients experienced disease progression. Chi-squared or Fisher’s exact tests were used to analyze the association between these two groups. There were no significant differences between the two groups in terms of age, sex, risk behaviors (including alcohol, betel, or cigarette consumption), primary site (buccal or non-buccal), grade, LVI, PNI, margin status, ENE, tumor stage (according to the 8th edition of the AJCC/UICC TNM staging system) [17], lymph node invasion, and pathological stage. However, more patients died in the disease-progressed group (19 patients, 52.8%) than in the progression-free group (7 patients, 10.6%), with significant differences (*p* = 0.001). Despite the insignificant findings for other characteristics, the disease-progressed group still shown a higher proportion in age ≥50 years (disease-progressed vs. progression-free: 77.8% vs. 62.1%) and experienced risk behaviors (91.7% vs. 89.4%). Moreover, the disease-progressed group also showed a higher proportion of poor clinical characteristics, including LVI (disease-progressed vs. progression-free: 13.9% vs. 7.6%), PNI (19.4% vs. 9.1%), ENE (13.9% vs. 6.1%), tumor stage IV (33.3% vs. 12.1%), lymph node invasion (27.8% vs. 24.2%), and advanced pathological stage (50.0% vs. 34.8%).

### 3.2. Unsupervised Hierarchical Clustering Analysis

Figure 2 demonstrates the analysis workflow of unsupervised hierarchical clustering analysis and the derivation of PRS using candidate protein clusters. In Step 1, the immunostaining scores of 16 target proteins were normalized to a staining matrix and visualized using a heatmap. In Step 2, the optimal number of clusters was eight, which was estimated using the silhouette index. In Step 3, the 16 target proteins were assigned to the eight protein clusters using the unsupervised hierarchical clustering method. In Step 4, the proportion of disease progression in each protein cluster according to the dichotomous risk group was computed.

The group with a higher proportion of disease progression was considered the high-risk group, while that with a lower proportion was considered the low-risk group. The red bar indicates the proportion of the high-risk group in disease progression, and the blue bar indicates the proportion of the low-risk group in disease progression. The details of the patient’s number, proportion, and log-rank test results of disease progression in the dichotomous risk group are summarized in Table 3. The protein clusters were ordered according to the number of target proteins, ranked from 1-factor to 4-factor. The number and proportion of disease-progressed in the high-risk and low-risk groups for each protein cluster were summarized, and the survival difference between both risk groups were estimated using the log-rank test. The optimal 1-factor protein clusters include p16 (high-risk vs. low-risk: 73.1% vs. 26.9%, log-rank *p* = 0.527), STK17A (80.8% vs. 19.2%, log-rank *p* = 0.677), and PIM1 (65.4% vs. 34.6%, log-rank *p* = 0.708). The optimal 2-factor protein clusters include EGFR–CDH3 (high-risk vs. low-risk: 96.2% vs. 3.8%, log-rank *p* = 0.151), KRAS–FLNA (73.1% vs. 26.9%, log-rank *p* = 0.205), and POLB–FEN1 (69.2% vs. 30.8%, log-rank *p* = 0.279). The optimal 3-factor protein cluster includes RB1–CDK6–CNSK1E (high-risk vs. low-risk: 69.2% vs. 30.8%, log-rank *p* = 0.745). The optimal 4-factor protein cluster includes PLK1–PhosphoMet–SGK2–SHC1 (high-risk vs. low-risk: 61.5% vs. 38.5%, log-rank *p* = 0.023). Survival analysis using the log-rank test indicated that only the 4-factor protein cluster had a significant survival difference between the high-and low-risk groups. Thus, the 4-factor protein cluster is selected as the candidate cluster for the PRS derivation shown in Step 5.

### 3.3. Cytoplasmic IHC Stainings and PRS Calculation

The PRS is generated based on the cytoplasmic IHC staining results of individual proteins included in the 4-factor protein cluster. The cytoplasmic IHC staining images of four individual proteins, including PLK1, PhosphoMet, SGK2, and SHC1, for high-risk and low-risk patients, were summarized in Figure 3. The PLK1, PhosphoMet, SGK2, and SHC1 IHC staining of tumor samples from high-risk (disease progression) patients mostly showed low cytoplasmic expression, while tumor samples from low-risk (progression-free) patients mostly showed high cytoplasmic expression. Low cytoplasmic expression includes negative to weak staining in IHC, and high cytoplasmic expression includes medium or strong staining in IHC. To fit the PRS calculation, we transformed the cytoplasmic expression into an immunostaining score ranging from 1 to 4, high to low cytoplasmic expression, respectively. For instance, the strong staining expression will inversely be transformed into score 1, and the negative staining expression will inversely be transformed into score 4.

The PRS calculation and risk strata identification of each patient were based on the transformed immunostaining score (*P*). The harmonic means of PLK1, PhosphoMet, SGK2 and SHC1 in the high-risk group were 2.46 (${\bar{H}}_{\mathrm{PLK}1}$), 3.08 (${\bar{H}}_{\mathrm{PhosphoMet}}$), 3.06 (${\bar{H}}_{\mathrm{SGK}2}$), and 3.21 (${\bar{H}}_{SHC1}$), respectively. According to Equation (1), the $S_{h}$ can be computed using Equation (4).
(4)$$S_{h}=\|P_{\mathrm{PLK}1}-2.46\|+\|P_{\mathrm{PhosphoMet}}-3.08\|+\|P_{SGK2}-3.06\|\;+\|P_{SHC1}-3.21\|$$

According to Equation (2), the $S_{l}$ can be computed using Equation (5). The harmonic means of PLK1, PhosphoMet, SGK2, and SHC1 in the low-risk group were 1.76 (${\bar{L}}_{\mathrm{PLK}1}$), 2.14 (${\bar{L}}_{\mathrm{PhosphoMet}}$), 2.38 (${\bar{L}}_{\mathrm{SGK}2}$), and 2.76 (${\bar{L}}_{SHC1}$), respectively.
(5)$$S_{l}=\|P_{\mathrm{PLK}1}-1.76\|+\|P_{\mathrm{PhosphoMet}}-2.14\|+\|P_{SGK2}-2.38\|\;+\|P_{SHC1}-2.76\|$$

Hence, the PRS can be computed using Equation (3), which is just simply subtracting the $S_{h}$ and $S_{l}$, and the positive PRS indicates the increased risk in disease-progressed, while the negative PRS indicates decreased risk in disease progression.

### 3.4. PRS Risk Strata Survival Analysis and Model Comparison

A total of 52 patients were considered and assigned to high-risk strata, and 50 patients were assigned into low-risk strata derived by PRS. Figure 4 illustrates the Kaplan–Meier plot of PFS and OS according to the PRS risk strata derived from the best multifactor protein combination. The 5-year PFS and OS rates of the high-risk strata were 49.0% and 65.6%, respectively. In addition, the 5-year PFS and OS rates of the low-risk strata were 67.6% and 77.4%, respectively. Compared with low-risk strata, the high-risk strata showed worse 5-year PFS (*p* = 0.023) and OS (*p* = 0.270).

Multiple Cox proportional hazard regression models were used to estimate the impact of PRS strata on PFS and OS in patients with oral cancer (Figure 5). Model 1 performed the univariate analysis including only PRS strata, and the results showed that PRS strata had a similar impact on PFS (HR = 2.20, 95% CI = 1.10–4.41, *p* = 0.026) and OS (HR = 1.55, 95% CI = 0.71–3.43, *p* = 0.274), but was only significant in PFS. Models 2 to 4 performed multivariate analyses, including PRS risk strata and multiple covariates. In model 2, demographic variables including age and sex were included as covariates, and the results showed that the PRS risk strata still had a significant impact on PFS (HR = 2.37, 95% CI = 1.17–4.83, *p* = 0.017) after model adjustment. Model 3 added the demographic variables and risk behaviors (any consumption of alcohol, betel, or cigarette) as covariates, while PRS risk strata still had a significant impact on PFS (HR = 2.35, 95% CI = 1.15–4.78, *p* = 0.019). Model 4 included the tumor characteristics site, grade, LVI, PNI, surgical margin status, ENE, tumor stage, lymph node invasion, and pathological stage. The adjusted model results also indicated that PRS risk strata still had a significant impact on PFS (HR = 2.47, 95% CI = 1.15–5.28, *p* = 0.020). Although PRS risk strata showed a similar impact on OS, no significant results were found in the adjusted Models 2 to 4.

## 4. Discussion

Oral cancer is a malignancy arising from the epithelium of the oral cavity, and the most common histology of oral cancer is squamous cell carcinoma. Due to its location in the oral cavity, oral cancer is more prone to exposure to external materials and carcinogens that construct a particular environment resulting in heterogeneous phenotypes. In the treatment of cancer, novel molecular identification and stratification have high clinical value because of their correlation with gene expression profiles, clinical features and phenotypes, overall survival, and prognosis. Gene expression profiling and its associated molecular expression have been widely used to generate a wealth of transcriptomic profiles in many cancer types [18]. Combinations of clinical parameters and gene-based biomarkers have also been used to predict prognosis and to evaluate therapeutic responses in other malignancies [19,20,21]. In our study, we performed unsupervised hierarchical analysis to stratify different patterns of patient groups and thus derived the progression risk score using candidate protein clusters, which enabled the generalization of the current study IHC findings into clinical settings. Furthermore, the PRS modules could be applied to survival analysis, including the Cox model, to investigate the simultaneous impact of baseline clinical characteristics and PRS risk on PFS.

The PRS can recognize the interaction between factors by considering associations within a protein cluster and is available in the extended analysis with clinical parameters in typical survival analysis approaches. The prognostic role of clinical parameters in oral cancer has been previously reported and indicated that oral cancer patients with a poorly differentiated grade, LVI, PNI, presence of tumor in the surgical margin, lymph node invasion, ENE, lymph node invasion, and the advanced pathological stage could obtain a poor survival outcome [22,23]. Moreover, the expression of four individual proteins, such as phosphoMet, was also altered in different clinical characteristics subgroups, including age and stage [24]. The study results demonstrated that PRS alone and incorporating clinical parameters, such as demographics, risk behaviors, and tumor characteristics, could be used as reasonable predictors of disease progression in patients with OSCC.

The PRS was generated based on a hierarchical agglomerative algorithm, and the avoidance of classification or dichotomous procedures in data preprocessing could partly prevent the loss of information and allow a decrease in type 1 errors (false positives). Consistent with previous studies of other cancers, we demonstrated that the agglomerative hierarchical clustering algorithm is advantageous for handling multifactor disease outcomes with uncertain interactions between multiple factors [25,26,27].

The four-factor protein cluster, including PLK1, PhosphoMet, SGK2, and SHC1, was found to be the optimal combination for predicting disease progression via IHC staining results. The PRS was estimated according to the co-expression of PLK1, phosphoMet, SGK2, and SHC1. Although PLK1 is frequently reported as an oncogene, the co-expression of PLK1 with specific genes could also play a tumor suppressor role and contribute to tumor progression by inhibiting the growth of oral cancer cells [28,29]. SGK2 is one of the isoforms of the SGK family, which is associated with the regulation of cell proliferation and survival [30]. The previous study has reported that inhibited SGK2 could induce cell death in multiple types of cancer, which could play a new role in cancer treatment [31]. SHC1, also known as p66Shc, is correlated to apoptosis-regulating gene expression [32]. The increasing expression of SHC1 could negatively regulate the T-cell activation and survival, which might result in poor survival outcomes in cancer patients [33,34]. PhosphoMet is commonly overexpressed in squamous cell carcinoma and involves increased gene copy number and mutation [35]. However, the phosphoMet could interact with multiple genes, such as HGF, PIK3, and SRC. The current study’s findings revealed converse results, which indicates the low cytoplasmic expression of PhosphoMet was associated with poor prognosis outcome and co-operated with low-expression of PLK1, SGK2, and SHC1. The function of PhosphoMet was complicated and was also affected by the clinical parameters of cancer patients. Hence, further evaluation for cytoplasmic alteration effects on the biological mechanism of PhosphoMet collaborating with PLK1, phosphoMet, and SHC1 should be investigated.

In summary, the simultaneous low expression of PLK1, phosphoMet, SGK2, and SHC1 might be associated with the immune response and tumor cell survival, which could result in poor survival outcomes in cancer progression. However, the results from IHC can vary widely depending on the staining protocol and the antibody or reagents used. The interpretation of staining patterns is another source of variability [36,37,38]. Thus, before any biomarker can be used clinically, it must be rigorously tested and validated in a large number of cases and in different laboratories. Recent advances in technology, such as the development of tissue microarrays (TMAs), should greatly help in this endeavor [39,40,41].

## 5. Conclusions

We identified a novel statistical method using unsupervised hierarchical analysis. By incorporating IHC biomarkers and clinical parameters, we found a potential biomarker, PRS, to predict PFS outcomes for OSCC. PRS is easily approached through IHC staining of surgical tissue specimens. Thus, we can stratify patients into high-risk groups and rapidly assess their outcomes based on the tumor expression of cytoplasmic PLK1, phosphoMet, SGK2, and SHC1. Hence, PRS may serve as a potential signature for predicting disease progression and assessing the risk of OSCC.

## Figures and Tables

**Figure 1 jpm-11-00908-f001:**
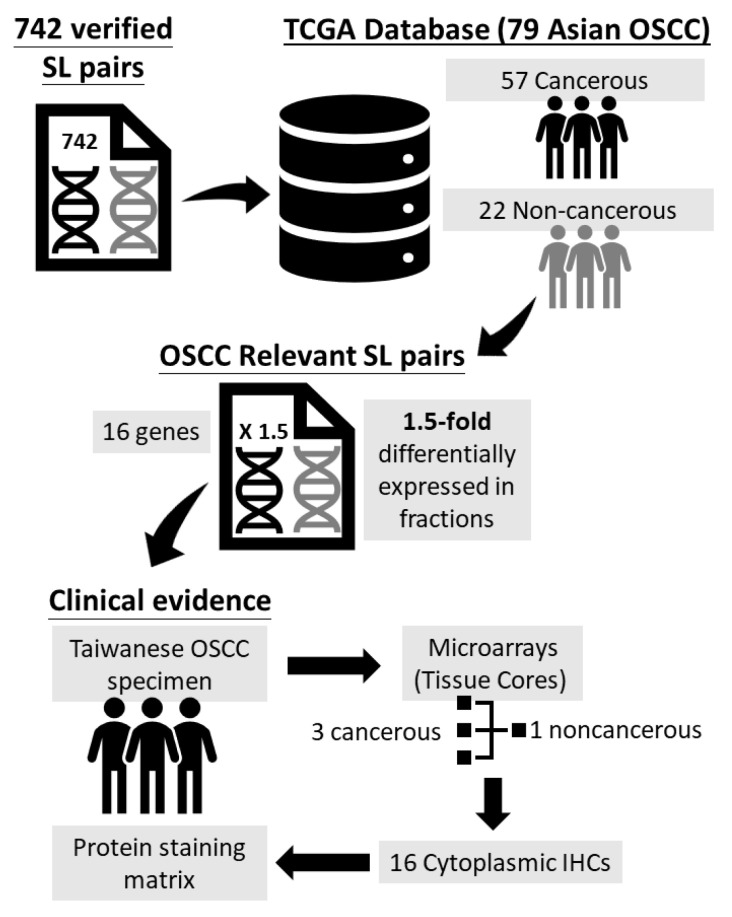
Schematic diagram for target gene selection from the validated synthetic lethality (SL) gene pairs and the identification of the protein staining matrix.

**Figure 2 jpm-11-00908-f002:**
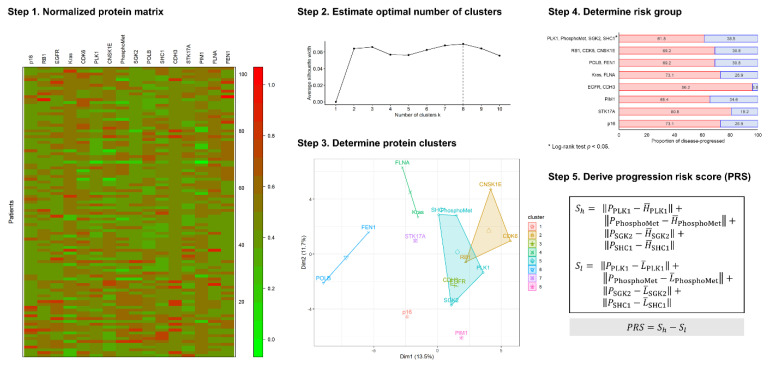
The analysis workflow for the protein staining matrix of target genes. (Step 1) Normalized the protein matrix of 16 individual cytoplasmic IHCs. (Step 2) Estimate the optimal number of clusters according to silhouette index. (Step 3) Determine protein clusters using the optimal number of clusters. (Step 4) Dichotomized and determined risk group according to the proportion of disease progression of each protein cluster. (Step 5) Derived progression risk score (PRS) using immunostaining score of the specific target protein within specific cluster, and the harmonic mean of the immunostaining score in correspond risk group.

**Figure 3 jpm-11-00908-f003:**
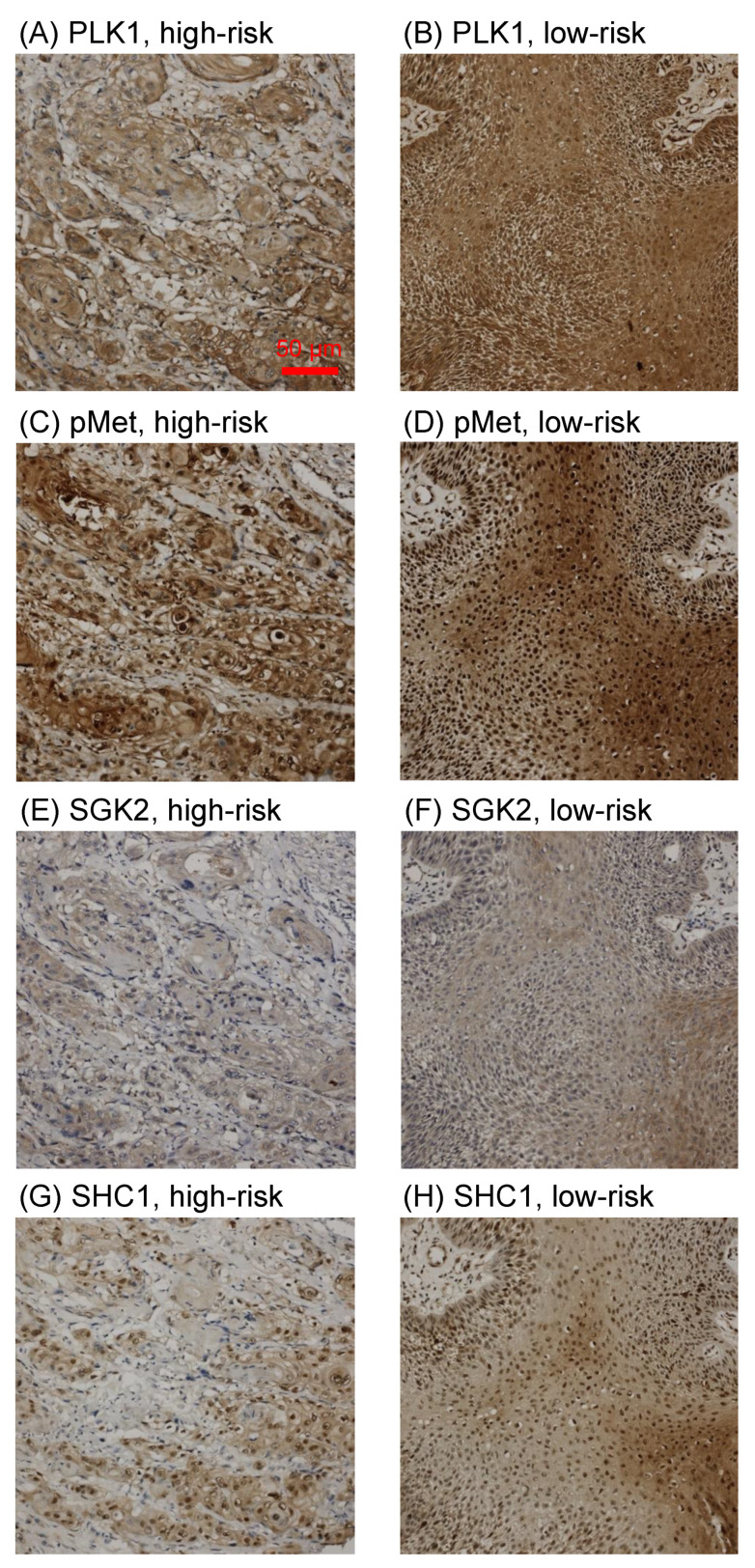
Tumor samples with IHC staining (magnification 400×) of 4-factor protein cluster, including (**A**,**B**) PLK1, (**C**,**D**) pMet: PhosphoMet, (**E**,**F**) SGK2, (**G**,**H**) SHC1, representing (**A**,**C**,**E**,**G**) low cytoplasmic expression in high-risk patients, and (**B**,**D**,**F**,**H**) high cytoplasmic expression in low-risk patients. The bar is 50 µm.

**Figure 4 jpm-11-00908-f004:**
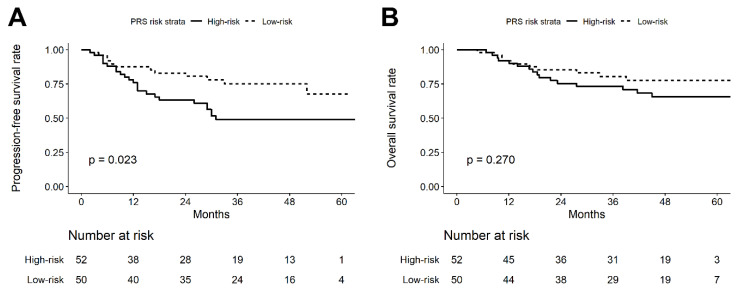
Kaplan–Meier plot of (**A**) progression-free survival (PFS) and (**B**) overall survival (OS) according to the PRS risk strata derived from the best multifactor protein combination.

**Figure 5 jpm-11-00908-f005:**
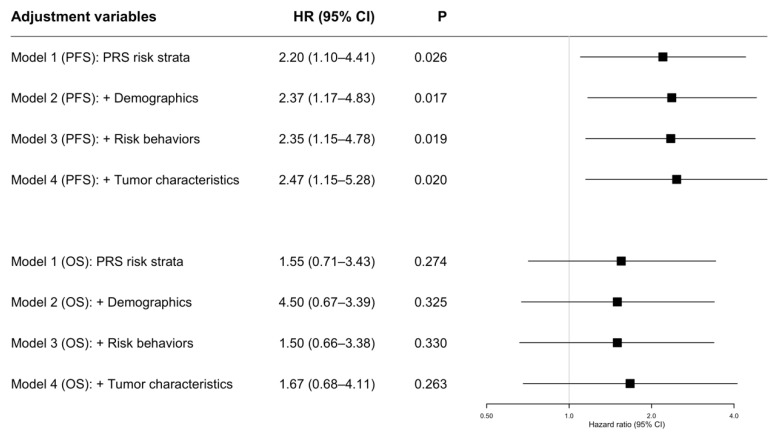
Estimation of the impact of progression risk score (PRS) strata on progression-free survival (PFS) and overall survival (OS) in oral cancer patients using Cox proportional hazard regression models.

**Table 1 jpm-11-00908-t001:** The antibodies and retrieval buffers for each protein.

Protein Name	Associated Protein Name	Clonality	Source	Catalogue Number	Dilution	Retrieval Buffer
CDH3	Cadherin 3	R	Abgent	AP1499B	1:50	T-EDTA
CDK6	Cell division protein kinase 6	R	Abcam Ltd.	ab124821	1:100	T-EDTA
CSNK1E	Casein Kinase 1 Epsilon	R	Abgent	AP7403a	1:50	T-EDTA
EGFR	Epidermal Growth Factor Receptor	R	Zeta Corporation	Z2037	1:50	T-EDTA
FEN1	Flap Structure-Specific Endonuclease 1	R	Abcam Ltd.	ab70815	1:1000	T-EDTA
FLNA	Filamin A	R	Abgent	AP7770a	1:50	T-EDTA
KRAS	KRAS Proto-Oncogene, GTPase Kirsten rat sarcoma virus protein	R	Abcam Ltd.	ab216890	1:200	C
MET ^a^	Mesenchymal epithelial transition factor	R	Abgent	AP3167a	1:50	C
P16	p16 (INK4a) tumor suppressor protein	M	BD biosciences	550834	1:100	T-EDTA
PIM1	Pim-1 Proto-Oncogene, Serine/Threonine Kinase	R	Abgent	AP7932d	1:50	T-EDTA
PLK1	Polo-like Kinase 1	R	Abgent	AP7937a	1:100	C
POLB	DNA Polymerase Beta	R	Abgent	AP50642	1:100	T-EDTA
RB1	Retinoblastoma 1	M	Leica Biosystems	NCL-L-RB-358	1:50	T-EDTA
SGK2	Serum/Glucocorticoid Regulated Kinase 2	R	Abgent	AP7947b	1:100	C
SHC1	Src homology 2 domain containing transforming protein 1	R	Abgent	AP50024	1:100	C
STK17A	Serine/threonine-protein kinase 17A	R	Abcam Ltd.	ab97530	1:100	C

^a^ PhosphoMet; R is Rabbit polyclonal; M is Mouse monoclonal; T-EDTA is Tris-EDTA buffer; C is Citrate buffer.

**Table 2 jpm-11-00908-t002:** Baseline characteristics of study population according to progression-free survival status.

Characteristics	Progression-Free	Disease-Progressed	*p*-Value
Cases	66	36	
Age			0.163
<50 years	25 (37.9%)	8 (22.2%)	
≧50 years	41 (62.1%)	28 (77.8%)	
Sex			1.000
Female	4 (6.1%)	2 (5.6%)	
Male	62 (93.9%)	34 (94.4%)	
Risk behaviors ^a^	59 (89.4%)	33 (91.7%)	0.984
Site			0.577
Non-buccal	29 (43.9%)	13 (36.1%)	
Buccal	37 (56.1%)	23 (63.9%)	
Grade			0.116
1	35 (53.0%)	13 (36.1%)	
2	29 (43.9%)	23 (63.9%)	
3	2 (3.0%)	-	
LVI	5 (7.6%)	5 (13.9%)	0.318
PNI	6 (9.1%)	7 (19.4%)	0.212
Margin not free	3 (4.5%)	3 (8.3%)	0.663
ENE	4 (6.1%)	5 (13.9%)	0.273
Tumor stage			0.055
I	32 (48.5%)	16 (44.4%)	
II	21 (31.8%)	6 (16.7%)	
III	5 (7.6%)	2 (5.6%)	
IV	8 (12.1%)	12 (33.3%)	
Lymph node invasion			0.878
Negative	50 (75.8%)	26 (72.2%)	
Positive	16 (24.2%)	10 (27.8%)	
Pathological stage			0.200
I-II	43 (65.2%)	18 (50.0%)	
III-IV	23 (34.8%)	18 (50.0%)	
Death	7 (10.6%)	19 (52.8%)	0.001

^a^ Risk behavior, including alcohol, betel or cigarette consumption. *p*-value was estimated using the chi-square test or Fisher’s exact test.

**Table 3 jpm-11-00908-t003:** The proportion of disease-progressed in each protein cluster derived by the unsupervised hierarchical clustering method.

Protein Clusters	High-Risk	n (%)	Low-Risk	n (%)	*p*-Value
1-factor					
p16	71	19 (73.1%)	31	7 (26.9%)	0.527
STK17A	77	21 (80.8%)	25	5 (19.2%)	0.677
PIM1	71	17 (65.4%)	31	9 (34.6%)	0.708
2-factor					
EGFR–CDH3	91	25 (96.2%)	11	1 (3.8%)	0.151
KRAS–FLNA	19	19 (73.1%)	83	7 (26.9%)	0.205
POLB–FEN1	66	18 (69.2%)	36	8 (30.8%)	0.279
3-factor					
RB1–CDK6–CNSK1E	73	18 (69.2%)	29	8 (30.8%)	0.745
4-factor					
PLK1–PhosphoMet–SGK2–SHC1	52	16 (61.5%)	50	10 (38.5%)	0.023

*p*-value was estimated using the log-rank test.

## Data Availability

The data presented in this study are available on request from the corresponding author, and we thank the Center for Research Resources and Development at Kaohsiung Medical University for their assistance in flow cytometry analysis.

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
