# Peer review of "Progression Risk Score Estimation Based on Immunostaining Data in Oral Cancer Using Unsupervised Hierarchical Clustering Analysis: A Retrospective Study in Taiwan"

_jpm, 2021, doi:10.3390/jpm11090908_

Round 1

Reviewer 1 Report

Comments to authors:
The authors attempted to establish a guideline for evaluation of oral cancer risk. It was found a novel statistical method using unsupervised hierarchical analysis in this study. Although these viewpoints are interesting, several points are ambiguous and difficult to interpret. The present work thus make it unlikely to be a convincing contribution to the field. The suggestions listed below may be considered for revision.

Abstract
1) ‘The progression-free survival (PFS) was significantly different between the two groups, and the high-risk patients were 2.20 times more likely to experience cancer progression than low-risk patients’ is difficult to interpret. Please rewrite.

Material and methods
2) The clinicopathological factors of patients enrolled this study should include ‘depth of invasion’, ‘recurrence’ and ‘blood vessel invasion’.
3) A table as a list of the clinicopathological factors should be presented. The authors stated ‘The baseline characteristics of the study population according to PFS status are summarized in Table 1.’ However I could not found the table.
4) The full spelling of 16 proteins should be indicated. Moreover, the immunohistochemical analysis is very important for this work. Thus, the technical contents regarding with immunohistochemistry (IHC) should not be showed as supplemental files.
5) The representative images of IHC should not be showed as supplemental files. In addition, the precise descriptions as figure legends are required.

Results
6) To verify the validity of the results from IHC, additional experiments using control IgG antibody as negative control are needed.

Discussion
7) What is ‘national guidelines in centralized settings ’?
8) The most contents of this section overlapped the Results section. Please overall rewrite.
9) I recommend to describe the information of PLK1, PhosphoMet, SGK2, and SHC1 using citation of  published papers.
10)  May I suggest the description about association of the 4 factors described above with aggressiveness of oral squamous cell carcinoma?

Reviewer 2 Report

In this manuscript the authors investigated a panel of IHC markers in a cohort of oral cancer (OSCC) patients, aimed to evaluate if the progression risk score obtained is useful for assessing prognosis of OSCC. In this retrospective case-control study, a cohort of 102 OSCC patients was included and a total of 16 markers were considered. A novel estimation approach with unsupervised hierarchical clustering analysis was used in this study. The results were interesting, showing that the prognostic risk score, based on a panel consisting of cytoplasmic PLK1, PhosphoMet, SGK2, and SHC1, successfully identified OSCCpatients with different progression-free survival.

The techniques utilized were appropriate and described with plenty details and the figures are helpful. Despite the small sample size, this is a well-designed study with rigorous methods. The discussion is balanced, and the statements are supported by the data. Overall, the work is interesting because gain in-depth analysis into prognostic biomarkers in OSCC. Indeed, the study is on a timely subject in view of increasing interest about the histological prognostic markers in oral cancer. There are few revisions needed before this work is suitable for publication:

  • In this study, the authors considered several morphological markers, such as LVI, PNI and, ENE. Therefore, in the Discussion section the authors could discuss the importance of the prognostic role of these morphological factors (e.g. regarding the prognostic role of pattern and localization of PNI in OSCC); for your convenience, DOI: 10.1111/odi.13900).
  • Minor: Only minor language corrections should be necessary.

Reviewer 3 Report

Summary

The manuscript with the title ‘Progression risk score estimation based on immunostaining data in oral cancer using unsupervised hierarchical clustering analysis: a retrospective study in Taiwan’ reported a study where 16 synthetic lethality (SL) associated genes were selected from the data downloaded from TCGA and GEO database, and then obtained the best combinations of 4 proteins from those 16 target genes from the unsupervised hierarchical cluster analysis of those 16 selected genes, the best 4-protein combination was therefore used to calculate the progression risk score (PRS). The authors claimed that PRS could serve as an ideal biomarker for clinical use in risk stratification and progression assessment in oral cancer. The authors did a good job in designing the study and introduced us a novel method and biomarker, PRS, but need to improve the manuscript organization and writing. One of my biggest concerns for the manuscript is the missing tables. Another issue in the manuscript is the method on how to calculate the PRS, the description and the formula are not clear and not enough for readers to follow their method in the manuscript to do the same work.

Comments

  1. The authors claimed they put the important data in two tables as they mentioned in the manuscript, Table 1 and Table 2. This is not true as there are no such tables found in the manuscript.
  2. The progression risk score (PRS) should be calculated for each patient who will be assessed for the progression risk for his/her oral cancer and then physician can determine the best intervention for the patient. It is the critical for readers to know how to exactly calculate the PRS introduced in the manuscript.

Question 1.

In Figure 2, Step 4, how to calculate the proportion of disease progression of each protein cluster and classify the patients into high/low risk group? More details are needed in either Figure 2 legend or the method section of the manuscript.

Question 2.

In Figure 2, Step 5, given a patient, how to calculate or obtain each P, H, L in the equations. The same questions apply to the equation 1, 2 and 3 on page5, see below.

On page 5, the authors presented three equations on how to calculate PRS but more details are needed to make it clear for readers. Given a patient, the authors need to explain in detail each symbol in each equation on page 5. What is the subscription ‘n’ in equation 1?  How to calculate or obtain ‘P1, …, Pn’ in equation 1? How to calculate or obtain ‘?̅n’? How to calculate ‘L1’ and ‘L2’ in equation 2? What is L1 and L2 in equation 2? Why is there no Ln in equation 2?

  1. In section 2.2. Tissue microarrays and Immunostaining, the last paragraph on page 3, ‘Tissue microarrays were further processed for the cytoplasmic IHC of 16 target proteins among the 21 genes.’. The authors need to explicitly provide readers the reasons or criteria why 5 genes were excluded and only 16 genes were selected as the biomarker genes in the study.

Round 2

Reviewer 1 Report

Is the revised supplementary figure S1 correct?

I found  the IHC images of control IgG antibody in your response note. However, the same images could not be found in the attached supplemental file.  Please check.

Author Response

Please check the supplementary Figure S1 in the attachment.

Reviewer 3 Report

Summary

The revised manuscript with the title ‘Progression risk score estimation based on immunostaining data in oral cancer using unsupervised hierarchical clustering analysis: a retrospective study in Taiwan’ showed some improvements in manuscript organization and writing, e.g., adding the missing tables in the revised manuscript but the authors need to be serious about the manuscript writing and presentation. The authors didn’t response and resolve all concerns I raised in the original manuscript. The authors’ response need to specifies how the authors addressed each comment I made. The response to my comments should be organized by presenting my comments and questions one by one, followed by the authors' response.

Comments

  1. The authors added the missing Table 2 and Table 3 in the revised manuscript but didn’t explain the details about the values in the tables. A detailed legend is necessary to ensure everybody know what values in table are.

For example, in the legend of Table 1, “Column Clonality: R is Rabbit polyclonal; M is Mouse monoclonal; ...”.

In the legend of Table 2, what is the column p? Is it actually the p-value? Please either use the more meaningful column name or explain in the legend what the column p is. Same issue exists in Table 3.

  1. The English writing needs improvement. For example, in Section 3.3, “Tumor samples for the high-risk patient (disease-progressed) were mostly obtained low cytoplasmic expression, and tumor sam-ples for low-risk patient (disease-free) were mostly obtained high cytoplasmic expression in PLK1, PhosphoMet, SGK2, and SHC1. The low cytoplasmic expression including neg-ative to weak staining, and the high cytoplasmic expression including moderate or strong staining.”.
